# A Geographical and Temporal Risk Evaluation Method for Red-Light Violations by Pedestrians at Signalized Intersections: Analysis and Results of Suzhou, China

**DOI:** 10.3390/ijerph192114420

**Published:** 2022-11-03

**Authors:** Jiping Xing, Qi Zhang, Qixiu Cheng, Zhenshan Zu

**Affiliations:** 1School of Transportation, Southeast University, Nanjing 211189, China; 2Department of Logistics and Maritime Studies, The Hong Kong Polytechnic University, Hung Hom, Hong Kong, China; 3Traffic Management Department of Suzhou Wujiang District Public Security Bureau, Suzhou 215299, China

**Keywords:** crosswalk, risk evaluation, geographical- and temporal-based risk, red-light pedestrian violation, signalized intersections

## Abstract

Red-light violations of pedestrians crossing at signal intersections is one of the key factors in pedestrian traffic accidents. Even though there are various studies on pedestrian behavior and pedestrian traffic conflicts, few focus on the risk of different crosswalks for the violating pedestrian group. Due to the spatio-temporal nature of violation risk, this study proposes a geographical and temporal risk evaluation method for pedestrian red-light violations, which combines actual survey and video acquisition. First, in the geographical-based risk evaluation, the pedestrian violation rate at signal intersections is investigated by Pearson correlation analysis to extract the significant influencing factors from traffic conditions, built environment, and crosswalk facilities. Second, in the temporal-based risk evaluation, the survival analysis method is developed to quantify the risk of pedestrian violation in different scenarios as time passes by. Finally, this study selects 16 typical signalized intersections in Suzhou, China, with 881 pedestrian crosswalk violations from a total size of 4586 pedestrians as survey cases. Results indicate that crossing distance, traffic volume on the crosswalk, red-light time, and crosswalk-type variables all contribute to the effect of pedestrian violation from a geographical perspective, and the installation of waiting refuge islands has the most significant impact. From the temporal perspective, the increases in red-light time, number of lanes, and traffic volume have a mitigating effect on the violations with pedestrian waiting time increases. This study aims to provide a development-oriented path by proposing an analytical framework that reconsiders geographical and temporal risk factors of violation. The findings could help transport planners understand the effect of pedestrian violation-related traffic risk and develop operational measures and crosswalk design schemes for controlling pedestrian violations occurring in local communities.

## 1. Introduction

Pedestrians are a vulnerable group of road users. Worldwide, the number of pedestrian deaths annually in road traffic accidents is about 270,000, which exceeds 22% of all traffic mortalities. This rate reveals its critical role in traffic safety research [1]. Of these, pedestrian accidents mainly occur at signalized intersections when pedestrians illegally pass-through crosswalks, which are highly dangerous due to the high risk of pedestrian–vehicle interactions. Illegal crossings mainly include pedestrians crossing at red lights or outside of marked crosswalks, and the former usually causes severe harm. This risky behavior may raise traffic safety issues between pedestrians and driving vehicles [2]. As such, it is necessary to analyze pedestrian red-light violations at signalized intersections and to reduce the risk of pedestrian crossing violations.

In this context, extensive efforts have been undertaken to investigate the influential factors of pedestrian crossing violations on red-light time. Among them, present studies can be divided into internal human factors and external environment by the differences in the purpose of the investigation. The internal human factors mainly consider the effects of age [3], gender [4], gap acceptance [5,6], mental effects [7], and crossing behavior selection as an individual or group of pedestrians [6]. External environment factors include built environment features [8], traffic conditions [9], the length of red-light time [10], time of the trip [11], social characteristics [12], and road crosswalk facilities [9,13]. It is a physical and mental decision-making process for pedestrians from the moment they arrive at a particular signalized intersection to the moment they are ready to violate the crossing. As the signal intersection is a complex traffic environment in urban transportation, the diversity of inherent personal characteristics and extrinsic intersection attributes can simultaneously affect pedestrian violations. This process has a high degree of uncertainty, and different pedestrians tend to use different crossing strategies under different crosswalk scenarios and waiting times.

However, most of the present studies on the risk evaluation of pedestrian violations have focused on the perspective of individual pedestrian characteristics and discussed the risk of illegal crossing from the perspective of pedestrians themselves. Fewer studies have discussed the risk of pedestrian crossing violations occurring from the effect of the entire crosswalk. Limited by the randomness and variability of pedestrians arriving at the crosswalk, measures to reduce the risk of violation from the pedestrian’s own perspective are uncertain. Instead, a whole crosswalk perspective can provide some suggestions for stable improvements based on the external environment. As such, it is crucial to consider geographical characteristics and temporal trends of selected external factors in the risk evaluation of pedestrian violations.

To solve this conundrum, this study presents a geographical and temporal risk evaluation framework for pedestrian violations at signal intersections. In the geographical-based analysis, the whole risk perspective of the signal intersection is evaluated by the violation rate of entries of the pedestrian group, and the influencing factors of traffic conditions, built environment, and crosswalk facilities are selected. In the temporal-based risk evaluation, the COX proportional-hazards model is developed to quantify the risk of pedestrian violations in different levels of external factors as time passes by.

The remainder of this paper is organized as follows. Section 2 proposes a literature review, the methodology is presented in Section 3, and Section 4 presents a case discussion. Finally, Section 5 concludes this paper and discusses outlooks for future research.

## 2. Literature Review

Since the 1990s, many researchers have focused on pedestrian safety. Many efforts have been made to determine and evaluate the factors that influence pedestrian violations and their severity. We divide the present studies on pedestrian violations into two parts: the studies on the geographical risk and those on the temporal risk of pedestrian violations.

### 2.1. Studies on the Geographical Risk of Pedestrian Violations

In the evaluation of geographical risk for pedestrians, pedestrian injuries are usually influenced by one specific factor or by the combination of several factors. In analyzing the risk affected by specific geographical factors, Nesoff et al. [14] discussed the geographic relationship between the spatial distribution of alcohol environments and pedestrian accidents in Baltimore City. Yao et al. [15] developed geographically weighted Poisson regression models for calculating the risk probability of a pedestrian collision when exposed to the roadway environment. Poulos et al. [16] introduced the relationship between pedestrian violations and the density of the surrounding buildings and population. Furthermore, Chaudhari et al. [17] explored the composition of vehicle types and the geometric linearity of the different road segments on the impact of pedestrian injuries. In the studies of pedestrian crossing affected by multiple geographical risk factors, Fuentes and Hernandez [18] discussed the relationship between macroscopic factors such as land use type, building density, and socioeconomic characteristics in urban pedestrian fatal accidents in Mexico. Jang et al. [19] compared the probability and the factors of pedestrian accidents in central business districts and urban suburbs. Rankavat and Tiwari [20] analyzed the impact of pedestrian accidents in different building environments and compared the risk levels of pedestrians in four different buildings. In different types of cities, Avinash et al. [21] found that the different sizes of cities lead to different crossing speeds in pedestrian violations. The average crossing speed was measured to be higher in the mega city (1.205 m/s) as compared to the metro city (1.036 m/s).

In studies of pedestrian violations affected by external factors, Alonso, Oviedo-Trespalacios, Gene-Morales, and Useche [13] found that the influence of age-based differences in pedestrian walking behavior is significant. Comparing different genders in pedestrian behavior, Useche et al. [22] introduced that male pedestrians are usually more sensitive than females in risk perception and error behavior observation. Ma, Lu, and Zhang [6] developed a Bayesian network-based framework for evaluating the influencing factors of illegal pedestrian behavior. Aghabayk, Esmailpour, Jafari, and Shiwakoti [10] identified the factors that affect illegal pedestrian behavior at signalized and un-signalized intersections and compared the behavioral differences between individual and group pedestrians with different ages and genders.

In general, previous studies on the geographical risk of pedestrian violations have been performed mainly from the perspective of the pedestrians themselves and analyzed the effect level of pedestrian accidents or fatalities with the help of macroscopic data (such as socio-economic development, demographic characteristics, and urban road network), or one specific factor’s data [23,24,25]. Different types of pedestrians present different crossing characteristics under different research scenarios in terms of travel patterns and traffic activities. However, fewer studies have focused on entire crosswalks at intersections to analyze the risk to pedestrians before an accident, which can help to propose improvements to mitigate the risk of pedestrian violations from the perspective of crosswalks. This topic is crucial to improve the safe environment at intersections.

### 2.2. Studies on the Temporal Risk of Pedestrian Violations

When pedestrians arrive at the intersection ready to cross, they will have a pre-judgment of violation decision during the waiting process. With pedestrian waiting times increasing, the probability of pedestrian violations also increases. As such, in this stage, the risk of pedestrian violations at the crosswalk is temporally variable. It is also essential to note the temporal change of pedestrian risk in this period. Tiwari et al. [26] examined the waiting duration of pedestrians at signalized intersections in India by a survival analysis method and found that the probability of risk significantly increases at the end of the waiting duration of red-light time.

Studying the effect via different pedestrian groups, Hamed [27] analyzed the relationship between the waiting time of pedestrians and the number of pedestrian groups. Liu, Alsaleh, and Sayed [11] compared the probability of violations for pedestrian groups under different numbers, different types of geographical locations, and different waiting times. Aghabayk, Esmailpour, Jafari, and Shiwakoti [10] examined the effects of different genders, ages, group crossings, and carry-ons on pedestrian crossing behavior or not, and found that distracted pedestrians had the greatest impact on pedestrian accidents.

In terms of different signal cycle lengths, Yang et al. [28] analyzed the relationship between pedestrian waiting time and crossing behavior. The study found that pedestrians are more inclined to end their waiting duration and engage in temporal violation behaviors as the waiting time increases. In a different cultural context, Sueur et al. [29] found that French pedestrians take more risks than Japanese pedestrians, and males take more risks than females, in the survival analysis model. The survival models have also found applications in the analysis of the reaction time of vehicle drivers. In a recent study by Pawar and Velaga [30], the driver’s reaction time (based on the response to an event) was examined with the help of a parametric survival model. The results suggested that the reaction of young drivers was 21% faster than that of mature drivers during the pedestrian crossing event.

In terms of external factors on pedestrian temporal risk, Dhoke et al. [31] developed a COX proportional risk model to analyze the joint effects of individual pedestrian characteristics and external environmental factors at intersections on pedestrian waiting times. Raoniar, Maqbool, Pathak, Chugh, and Maurya [12] pointed out that regardless of whether the crosswalk is signal-controlled or not, the waiting time for crossing should not exceed 50 s, otherwise the probability of pedestrians crossing illegally will increase.

In summary, research can benefit from pedestrian waiting time data collected utilizing interviews, questionnaires, video trajectory recording, etc. [12,32,33]. These studies have pointed out the maximum waiting times for pedestrian crosswalk violations and described the changing trends in the risk of pedestrians, from pedestrians arriving at crosswalks to the occurrence of the violation. However, the aforementioned studies did not reflect the effects of various external factors on the risk of the pedestrian violations involved. The impact of different levels of external factors on the temporal risk of pedestrian violations is of different degrees. For example, the waiting time for pedestrians under different crossing facilities is different.

Hence, the contributions of this study are two-fold. First, a geographical and temporal risk evaluation framework for evaluating the whole process of crosswalk violations is proposed. Second, the violation rates from different intersections are viewed as the standard that affects the violation of crosswalks, and we collected data from 16 real intersections in Suzhou, China, for statistical analysis.

## 3. Methodology

This study adopted a combination of video-based acquisition and survey confirmation methods to obtain data on pedestrian violations at intersections. The Pearson correlation analysis and survival analysis methods were applied to evaluate geographical- and temporal-based pedestrian violation risk, respectively.

### 3.1. Research Framework

To achieve a reliable evaluation result, the main steps are described as follows:

Step 1: Typical signal intersections were selected in Suzhou, China. The data acquisition method and sample size of pedestrian violations were determined. Pedestrian characteristics, traffic conditions, built environment, and crosswalk facilities were collected from each signal intersection.

Step 2: The significant influencing factors of intersection violation rates were determined by Pearson correlation analysis, and a risk evaluation model of the risk of the crosswalk at each intersection was established.

Step 3: A COX proportional-hazards model for risk evaluation was developed to identify the influencing factors of pedestrian waiting time. The effect of each factor on pedestrian waiting time at different risk levels was described in pedestrian crosswalk violations. The research framework is shown in Figure 1.

### 3.2. Investigation of Pedestrian Crosswalk Violations

#### 3.2.1. The Selection of Study Area

Due to the complexity of urban traffic conditions and the diversity and uncertainty of pedestrian behaviors, the selection of intersections for investigation will directly affect the analysis of pedestrian crosswalk violations [34,35,36,37]. For the purpose of this study, this investigation is limited to the crosswalks of signal intersections, and it is necessary to investigate the pedestrian characteristics and environmental factors of external road traffic.

We selected the violation rate at intersections as our object to compare the impacts from different intersections. The datasets of pedestrian violations from multiple intersections were collected. To be efficient, we chose a combination of video and survey collection. First, we considered the characteristics of crosswalks at different intersections. Due to the different characteristics of the location of crosswalks, pedestrians have different crossing decisions. Pedestrians located in commercial districts are driven by the efficiency requirement from work, they prefer to cross the street with impatience, and the probability of crossing violations is high. In contrast, pedestrians in residential areas are mainly in demand for leisure, and thus the probability of crossing violations is low. Therefore, the selection of intersections for investigation included crosswalks in different functional areas.

Second, the signal intersections also met the following requirements: (1) The crossing pedestrian flow should have a large enough sample size and be close to some traffic attraction areas. (2) It is required that there is no obvious obstacle within the scope of the investigation. (3) The types of crosswalk facilities should include three types of crosswalk facilities: one-time crosswalk, mid-block crosswalk, and crosswalk with a waiting refuge island, as shown in Figure 2.

In terms of investigation time selection, to ensure high-quality investigation data, peak traffic periods were selected [38,39,40]. Furthermore, the investigation time was during the working day and daytime, and the weather was sunny. The investigation periods were: morning peak (7:00–9:30 a.m.), afternoon peak (11:30 a.m.–12:30 p.m.), and evening peak (5:30–7:00 p.m.). The study duration at each intersection was at least 30 min. With the aforementioned investigation requirements, 16 typical street crossing sites were selected for investigation, and the description is listed in Appendix A. The video surveillance data of the investigation were provided by the Suzhou Traffic Administration Department.

#### 3.2.2. Determination of Risk Factors

In the study of risk factors of a pedestrian crossing from a geographical and temporal perspective, we selected the ratio of violating pedestrians to the whole pedestrian group at the intersection as a geographic criterion and the length of pedestrian waiting time as a temporal criterion, and the video-based data acquisition method was applied. We captured images from the video to count the waiting time of pedestrians and whether a violation occurred. The process detail of the surveillance video is shown in Figure 3. The left figure indicates the recording process of pedestrian waiting time statistics and the right figure represents the statistics of traffic volume.

By viewing a pedestrian as a unit, pedestrian waiting time data were obtained by video recording the time between the arrival of the pedestrian at the intersection and the start of the crossing. When a pedestrian crosses a crosswalk at the red-light time, it is considered a crossing violation [41]. The data on waiting times for pedestrian violations are publicly available on GitHub (https://github.com/xingjiping/Data-of-waiting-time-for-pedestrian-violation (26 September 2022)). Furthermore, in the study of risk factors of pedestrians crossing illegally, whether the crosswalk facilities are reasonable or not directly influenced the comfort of pedestrians using these facilities, thus indirectly influencing the risk of pedestrians crossing the street illegally. The traffic state at signal intersections usually directly affects the crossable gap for pedestrians [39,42], and the built environment around the intersection determines the composition of the pedestrian crossing group [7,43]. We selected traffic conditions, the built environment, and crosswalk facilities of crosswalks as the main categories of our investigation.

These risk factors are listed in Table 1. It should be noted that in the division of land use types, based on the percentage of building types within a 100 m radius around the crosswalk at an intersection, land use types around the crosswalk can be divided into the residential area, commercial area, and mixed area. For example, if a supermarket or residential community appears near the intersection, it can be classified as a commercial area or residential area, respectively, otherwise, it is regarded as a mixed zone [19,20]. As for the following modeling, these factors were defined as different categories of variables by their properties, and then the variables were assigned values. Furthermore, for different types of pedestrians, we further classified them into different ages and genders based on their physical features in the video.

### 3.3. Pearson Correlation Analysis of Violation Factors

In the geographic risk evaluation, we took the crosswalk at each intersection as our research object and evaluated the impact of geographical environmental factors on the violation rate at this crosswalk. Then, we filtered out the significant influencing factors by Pearson correlation analysis. In pedestrian crash estimation, the Pearson correlation coefficient can be used to judge whether the variables changed along a trajectory, and then to determine the linear relationship between variables of a fixed distance type [44]. Before correlation analysis, the scatter plot can be drawn first, then the hypothesis of the correlation coefficient can be tested, the correlation coefficient between variables can be calculated, and the relationship between related variables can be described. Therefore, Pearson correlation analysis is widely used to judge whether there is a correlation or not.

The correlation between variables was measured by calculating the Pearson correlation coefficient, and the Pearson correlation coefficient can be calculated as follows:(1)$$\gamma_{j}=\frac{\sum_{i=1}^{n}\left(x_{ij}-{\bar{x}}_{j}\right)\left(y_{i}-\bar{y}\right)}{\sqrt{\sum_{i=1}^{n}{\left(x_{ij}-{\bar{x}}_{j}\right)}^{2}{\left(y_{i}-\bar{y}\right)}^{2}}}=\frac{\sigma_{x_{j}y}^{2}}{\sigma_{x_{j}}\sigma_{y}}$$
where n is the number of signal intersections, the variable $x_{ij}$ denotes the selected jth influencing factors at signal intersections i, the variable $y_{i}$ represents the pedestrian violation rate at signal intersections i, i=1,⋯,n, $\sigma_{x_{j}y}^{2}$ is the covariance of variables, and $\sigma_{x_{j}}$ and $\sigma_{y}$ are the standard deviations, respectively. When $\gamma_{j}>0$, $x_{j}$ and y are positively correlated. When $\gamma_{j}<0$, $x_{j}$ and y are negatively correlated. When $\gamma_{j}=0$, there is no correlation.

The greater the absolute value of the Pearson correlation coefficient, the stronger the correlation. Generally, the correlation of variables is judged by the value ranges. As such, if $\left|\gamma_{j}\right|\leq 0.3$, there is no linear correlation, if $0.3<\left|\gamma_{j}\right|\leq 0.5$, there is a low linear correlation, if $0.5<\left|\gamma_{j}\right|\leq 0.8$, there is a significant linear correlation, and if $0.8<\left|\gamma_{j}\right|\leq 1$, there is a high linear correlation [45].

In this section, a multiple linear regression model is developed for modeling pedestrian violation rates affected by different factors in intersection geography, which is based on regarding the pedestrian violation rate as the dependent variable and each selected significant factor as the independent variables. The geographical-based pedestrian crosswalk risk evaluation model can be formulated as follows:(2)PedVR=α1⋅W+α2⋅Q+α3⋅Tred+α4⋅CI+C
where PedVR represents the pedestrian violation rate,  C denotes constants, W represents the crossing distance (m), Q is traffic volume at the crosswalk (Pcu/5 min), Tred denotes the length of red-light time (s), CI is the crosswalk type (one-time crosswalk, mid-block crosswalk, and crosswalk with a waiting refuge island are set as 1, 2, and 3, respectively), and α1,α2,α3,α4 represent regression coefficients.

### 3.4. Survival Analysis of Pedestrian Waiting

In this section, we select the violation probability while waiting to cross the street illegally as the indicator of the temporal risk of pedestrian violations. When pedestrians arrive at an intersection and begin to wait to cross, as the waiting time increases, the pedestrians will become impatient, the probability of violation increases, and the risk to pedestrians correspondingly increases.

With the help of pedestrian waiting time, the survival analysis method was applied to evaluate the temporal risk of pedestrian crosswalks. Survival analysis is used to analyze the survival time of events by processing the lifecycle of data that include the internal and external risk factors, and it is mainly used to calculate the probability that events can continue to survive, which includes the survival function, probability density function, and danger probability function [26].

In this section, the process of waiting for a crosswalk is regarded as a survival process, and the period from waiting to leaving the intersection is viewed as the survival time. Herein, to analyze the influence of individual factors on the risk of violation, we used the Kaplan–Meier method to describe the survival curve and the risk curve of a pedestrian during the waiting process. Then, to evaluate the impact of multiple factors collectively on the risk of violation, we applied a COX proportional-hazards regression method to evaluate the risk degree of pedestrian violations, which is influenced by different risk factors.

The survival function reflects the probability that an individual survives to time t (experiencing an event (whether a violation or not) after time t). The survival function of pedestrian waiting is the proportion of pedestrians who continue to wait after waiting for time t [46]. Among them, s(t) is the survival function and f(t) represents the probability density function. The survival function is as follows:(3)$$s(t)=P(T>t)=\int_{t}^{\infty }f(t)$$
where P(T>t) denotes the probability of the violation occurring when the time {T>t}. Since the survival function is the integral of the probability density function f(t), it is recorded as:(4)$$f(t)=\underset{\Delta t\to 0}{\lim}\frac{p(T\leq t+\Delta t)}{\Delta t\to 0}=\frac{dS(t)}{dt}$$
where the probability density function, f(t), is the change of the survival rate of pedestrians crossing the street after waiting for a certain time Δt, and the area between the curve of the f(t) function and the coordinate axis is 1.

In our proposed method, the data from pedestrians who arrived at the red-light time and crossed the street during the green-light time in the next cycle were also useful. In judging whether pedestrian violations occurred or not, the pedestrian waiting time data can be divided into complete data and censored data. The complete data are the waiting time data of pedestrians from when they arrived at the crosswalk to crossing illegally, and the censored data are the waiting times to when pedestrians normally crossed the street, which is not risky. In the censored data, even if the waiting time of pedestrians did not reach the maximum waiting time, it still contained waiting time information and is necessary together with complete data for evaluating the risk of pedestrian waiting.

In the description of survival and hazard curves, we selected the Kaplan–Meier method for our research sample size to estimate the survival rate. The Kaplan–Meier estimator is a non-parametric maximum likelihood estimator of the survival function [47]. It is a piecewise constant, which can be thought of as an empirical survival function for censored data and yields an unbiased estimate of survival.

Let k be the number of pedestrians in the intersection i, and $T_{1}$, $T_{2}$, …, $T_{k}$ denote the waiting times of pedestrians in the crossing violation group, and they are viewed as complete data. $C_{1}$, $C_{2}$, …, $C_{k}$ are the waiting times of pedestrians in the normal crossing group, and they denote censored data. Both have the same survival function, s(t), and they can be defined as:(5)$$s_{n}(t)={\prod_{Y(i)\leq t}\left(1-\frac{1}{n-i+1}\right)}^{\delta (i)}$$
(6)$$Y_{i}=\min(T_{i},C_{i})$$
(7)$$\delta_{i}=IT_{i}<C_{i}$$
where δi denotes the truncate function: if Ti≤Ci, it represents complete data, δi=1, and if Ti>Ci, it is censored data, δi=0.

After plotting the survival and hazard function curves of pedestrians impacted by each risk factor, based on the application of the COX proportional-hazards model, we further analyzed the degree of risk of pedestrian violations affected by the combination of different risk factors during the waiting period. In other words, the purpose of the COX model is to simultaneously evaluate the effect of several factors on survival [11,48]. It can quantify how specified factors influence the hazard rate of pedestrian violations at a particular intersection in a time slot. These factors can be regarded as covariates in the survival analysis.

The COX model can be interpreted as the risk of violation at a time t, which can be estimated as follows:(8)$$h(t|x)=h_{0}(t)exp(\sum_{j=1}^{P}a_{j}x_{j})$$
where h(t|x) is the instantaneous hazard rate at the time t for a covariate X=x, h0(t) denotes the baseline hazard function, and $a=(a_{1},a_{2},\ldots a_{p})$ is a vector of different regression coefficients. The magnitude of the risk of an individual pedestrian at the time t in this model was determined by two sub-models. One is the baseline risk, $h_{0}(t)$, which represents the risk of violation without any influence of factors in time t, and the other is individual specificity, $\exp(\sum_{i=1}^{p}a_{i}x_{i})$, which is the increased risk from the i-th factor at the time t.

## 4. Case Study

### 4.1. Statistics of Pedestrian Violation Data

With the sample size of pedestrian violations and the total pedestrian statistics from the captured video, we obtained 4589 pedestrian crossing data points at signal intersections, which included 881 crossing violation pedestrians and 3708 normal crossing pedestrians. Hence, the ratio of crossing violations is the ratio of the number of pedestrian violations to the total number. We selected one of the crosswalks at each signalized intersection as our research objective to analyze the pedestrian violation rate. Before conducting our work, we verified the reliability and validity of the data, and justified the data through discussions with experienced investigators. Furthermore, the sites of 16 typical signalized intersections in Suzhou, China, were determined, and the details of the external facility descriptions for each intersection are listed in Appendix B. The total violation rate for each intersection is shown in Figure 4.

### 4.2. The Identification of Geographical Risk Factors of Violation

As mentioned in Section 3.2.2, we initially determined 10 factors from the traffic conditions, built environment, and crosswalk facilities as geographical factors influencing pedestrian violations at each signal intersection. With the survey and video recording statistics, we further validated the significance of each risk factor by pedestrian violation rates. In this study, Pearson correlation analysis was applied to identify the significance of risk factors. The correlation coefficients between risk factor variables and violation rates were calculated by SPSS statistical software (SPSS Inc., Chicago, IL, USA). Among the nominal variables in the independent variables, we performed correlation analysis in numerical form, and the results are shown in Table 2.

According to the requirement of the t-value significance test, the t-value of significant influencing factors should be less than 0.05 [49]. Table 2 shows that the four factors satisfy this significance level. It includes the crossing distance, traffic volume, red-light time, and type of crosswalk. This demonstrates a significant correlation between them and pedestrian violation rates. Furthermore, the values of significance in the other 6 factors (number of lanes, crossing distance, crosswalk isolation degree, whether to set red-light countdown, land use type around crosswalk, signal clearing time) all exceed 0.05, and these factors are not significantly correlated. Thus, we find that pedestrian violations are related to the crossing distance, the length of red-light time, the traffic volume, and the types of crosswalks.

### 4.3. The Geographical Risk Model for Pedestrian Violations

Based on the violation data at the 16 selected signal intersections, we obtained the regression coefficients and t-values in the model by SPSS statistical software, as shown in Table 3.

In Table 3, the positive and negative coefficients were used to identify the role of each factor on the violation rate. Among them, crosswalk distance was negatively correlated with violation rate, which indicates that too-long crossing distance causes the violation rate to be low, and this is consistent with the findings in [6]. Similarly, when the traffic volume at crosswalks is high, it caused a decrease in the probability of violations. In contrast, when the red-light time was longer, the pedestrians showed impatience, preferring to violate, and similar discussions can be found in [5,9]. Comparing the previous studies, we also evaluated the violation rate in three different types of crosswalks and found that the safety at the one-time crosswalk, mid-block crosswalk, and crosswalk with waiting refuge island is gradually decreasing. Furthermore, the size of the value represents the level of impact, and the crosswalk type created the greatest impact, which brings the most significant improvement in crossing safety. The second effect was traffic volume, where the higher the traffic volume, the more cautious the pedestrians usually are at the crosswalk.

### 4.4. The Temporal Risk Factors for Pedestrians Waiting at the Crosswalk

In the process of pedestrians waiting at the crossing, pedestrians are affected by waiting times, the probability of violation varies, and thus the risk of pedestrian crossing fluctuates [50,51]. Combining the aforementioned geographical risk factors that are significant in pedestrian violations, we continued to utilize part of the selected factors to analyze pedestrian temporal risk, which includes red-light time, traffic volume, number of lanes, and crosswalk type. Furthermore, combining the factors selected by similar studies, the factors of age, gender, and the number of lanes also cannot be neglected [11].

Hence, age, gender, red-light time, traffic volume, number of lanes, and crosswalk type are considered initial temporal risk factors. To compare the effects of the three factors at different levels for the risk of violation, we divided them into different intervals. The red-light time was set in 3 time slots, which include 40–80, 80–100, and 100–150 s. The traffic volume could be divided into three ranges of 0–1000, 1000–2000, and 2000–5000 pcu/h, and the crosswalks with 4–6, 7–8, and 9–10 lanes were grouped [52,53]. The crosswalk types include one-time crosswalks, mid-block crosswalks, and crosswalks with a waiting refuge island. In the representation of age and gender, teenager = 1, adult = 2, elderly = 3, and male and female were denoted as 1 and 2, respectively.

Then, we also needed to verify whether these factors play a critical role in our pedestrian waiting time experiments. To avoid the omission of potentially valuable variables, the backward elimination method was applied to gradually remove the insignificant covariates [54]. As shown in Table 4, since the factors of age, gender, and crosswalk distance were not significant, they were eliminated in the first, second, and third steps, respectively. Note that the number of lanes does not equate to the crosswalk distance, which is dependent on whether the crosswalk is set up with crossing refuge safety islands and road central barriers or not. Unlike in the geographical risk analysis, which is influenced by the violation rate of crosswalks, there were some co-linearity problems between each influence factor. The factor analysis of the temporal risk took the violation waiting time of each individual pedestrian as the dependent variable and analyzed the variation of waiting times of different pedestrians in different external environments. Therefore, it is acceptable that there were differences in the factors extracted for geographic and temporal risk analysis.

Herein, the Kaplan–Meier method was used to estimate the variations in the risk of violation that were affected by each factor as pedestrian waiting time passes by. Considering the effects of censored data and non-censored data, the curves of the survival and hazard functions of pedestrians with these four factors are plotted in Figure 5 and Figure 6.

The survival and hazard function curves by the length of red-light time are shown in Figure 5, where we found that the survival rate dropped rapidly in the first 10 s of waiting time. This reflects that the proportion of pedestrians who were ready to violate in this period was high. Pedestrian violation is usually concentrated in the early stage of the red light and at the end of the green light, when vehicles start to pass the crosswalk, or vehicles finish their crossing at the beginning or end of the green-light stage. At this signal transition phase, there will be gaps in the crosswalk. Furthermore, with the same waiting time, the crosswalk with a red-light time in the 40–80 s range had the lowest survival rate and the highest risk rate. For example, in the 20 s waiting time, the survival probability of pedestrian violations in the 40–80 s range was almost 20%, while this probability was about 30% in the 80–100 s range, and 40% in the 100–150 s range. Hence, the longer the red-light time, the higher the survival rate of pedestrian violations while waiting for the same time.

Different from [11] in the discussion of the effect of the number of pedestrians, crossing at different times, and the land use type of crosswalk waiting time, we focused mainly on the impact of the external environment of crosswalks at the intersection. With the effect of different levels of traffic volume, the three survival function curves by traffic volume all showed a decreasing trend in the pedestrian survival rate. The survival rate effect by traffic volume in the 0–1000 pcu/h range was smaller than that in the 1000–2000 and 2000–5000 pcu/h ranges, and gradually rose with increased traffic flow. At the same survival rate, the proportion of pedestrian violations was the greatest when the traffic volume was 0–1000 pcu/h. This indicates that the proportion of pedestrian violations increased when traffic volume was small and the gap in the crosswalk was large, which is consistent with the perception that traveling vehicles will serve as a warning to pedestrians in their decision-making [55]. With the increase in traffic volume, the survival rate of pedestrian violations became high, the proportion of pedestrian violations was low, and the tolerable waiting time increased.

As shown in the survival function and hazard function curves by the different numbers of lanes in Figure 6, the waiting times for the pedestrians were also different due to the effect of the number of lanes. There was a positive correlation between waiting time and the number of lanes, whereby the more lanes there were, the longer the waiting time for the pedestrian violation, and the higher the risk. When the number of lanes was 4–6, the inflection point of the survival curve appeared in a 37 s waiting time. When the waiting time was 50 s, the survival rate reached 0. The inflection point in the waiting time was 45 s when the number of lanes was 7–8. Similarly, the waiting time of the inflection point was 60 s when the number of lanes was 9–10. When the waiting time was about 80 s, the survival rate reached 0. Compared with the related research conducted in India [12,56] and the Arabian region [9], similar findings can be acquired.

As shown in the survival function curve effect by different crosswalk types in Figure 6, the waiting time for pedestrian violations gradually increased when the crosswalk type switched from the one-time crosswalks, mid-block crosswalks, to crosswalks with a waiting refuge island. Among them, the survival rate of the maximum waiting time of one-time crosswalks approached 0 at 40 s, reached 0 at the mid-block crosswalks at 60 s, and reached 0 again at waiting time of 80 s in the crosswalks with a waiting refuge island.

### 4.5. The Temporal Risk Model for Pedestrian Violations

After the Kaplan–Meier-based method for temporal risk, each factor has been analyzed on violation waiting time. It is also necessary to analyze the risk of violations under the joint influence of multiple factors. In this section, the COX proportional-hazards model is applied for quantifying the impact of multiple factors.

Based on the SPSS statistical software, the regression coefficients of the model were obtained in Table 4, and our proposed COX proportional-hazards model can be expressed as Equation (9):(9)h(t)=h0(t)exp(0.144Lane−0.183Redtime−0.120Crosstype−0.434Trafficvolume)
where *t* is the survival time, and it denotes the waiting time before pedestrian violations, $h_{0}(t)$ denotes the basis hazard function and it is affected by external uncertainties, such as weather, local travel culture, etc. [54]. h(t) represents the calculated risk value for pedestrian violations in time *t*. Furthermore, the variable lane denotes the number of lanes, the variable *redtime* is the length of the red-light signal time, *crosstype* represents the value of different crosswalk types, and *trafficvolume* is the value of the classified traffic volume category.

Due to the uncertainty of $h_{0}(t)$, it is a nonparametric model. We applied a prognostic index (*PI*) to replace the calculation of fixed risk. The *PI* for the temporal risk of pedestrians is as follows:(10)PI=0.144Lane−0.183Redtime−0.120Crosstype−0.434Trafficvolume

In Equation (10), the variable coefficient reflects the degree of influence of each risk factor. When the coefficient is positive, it means that the hazard function decreased with the increase of the level of the influencing factor. When the coefficient is negative, it means that the risk of pedestrian violations increased with the increase of the level of the influencing factor. For example, in the specific traffic scenario of a crosswalk at an intersection, the value of *PI* can be obtained by inputting the number of lanes, with the categories of red-light time, traffic volume, and crosswalk type. With a determined basis hazard, $h_{0}(t)$, the temporal risk of pedestrian violations in time t can be acquired by h(t).

## 5. Conclusions

The study contributes to understanding the joint effect of traffic conditions, built environment, and crosswalk facilities on the pedestrian crossing violations, which could help decision-makers to analyze comprehensive countermeasures to decrease the risk of violations. From a total of 16 signal intersections in Suzhou, China, over 4500 samples of pedestrian crossings were collected. Then, this study proposed a new geographical and temporal evaluation method to examine the effects of these factors on the risk of crossing violations. Traffic volume, crosswalk types, red-light time, and crosswalk distance were identified as main factors by Pearson correlation analysis with the pedestrian violation rate. With an increased pedestrian waiting time, the temporal impact of each factor was described by survival function and hazard function curves, which were based on the Kaplan–Meier method. The COX proportional-hazards model for each indicator was used for comparing the degree of the impact of different factors on the risk of pedestrian violations.

In the geographical risk evaluation, the crosswalk type caused the greatest degree of effect in the four selected significant influencing factors. This indicates that the installation of crossing refuge islands can significantly decrease the crosswalk risk at signalized intersections. Furthermore, the increase in the values of traffic volume and crossing distance conversely led to a decrease in the number of pedestrian violations. The length of the red-light time had the opposite effect on these two items. In the temporal risk analysis, we found that the risk of pedestrian violations increased as waiting time increased, regardless of the significant influencing factors.

Based on our findings in the geographic and temporal risk analysis, for policy suggestions on improving the safety of pedestrians crossing at intersections, crosswalk planners should ideally install crossing refuge islands within the limitations of land use. The set length of red-light time needs to be comprehensively integrated into different external environments.

## 6. Limitations and Further Research

Although this study used a sample that was representative of pedestrians and the essential statistical parameters, and theoretical assumptions were coherently achieved during the data analysis, there are still some limitations and potential biasing sources in the collection of violation waiting time data that must be acknowledged. Particularly, the judgment of whether a violation has occurred after a period of pedestrian waiting, and the recording of the start time of multiple pedestrians arriving at a crosswalk simultaneously. However, this does not inhibit the influence of common method biases and variance, mainly if sensitive issues related to age, gender, and violation criteria from pedestrians are addressed, and these do not prejudice our discussion of the external risk factors for crosswalks.

In future research, with the help of the questionnaire, we could further extend the crosswalk perspective to investigate the effect of physical characteristics, trip purpose, trip distance, and trip time of pedestrians. The evaluation of violation risk can be further analyzed by spatial geographical location, which includes the degree of impact on violation by different spatial distances between the intersection and between each intersection and the diversity of the built environment. Moreover, our study only utilized data from 16 signalized intersections, and consideration can be taken to increase the number of investigated intersections to comprehensively evaluate the relationship between the influencing factors in a large-scale urban road network.

## Figures and Tables

**Figure 1 ijerph-19-14420-f001:**
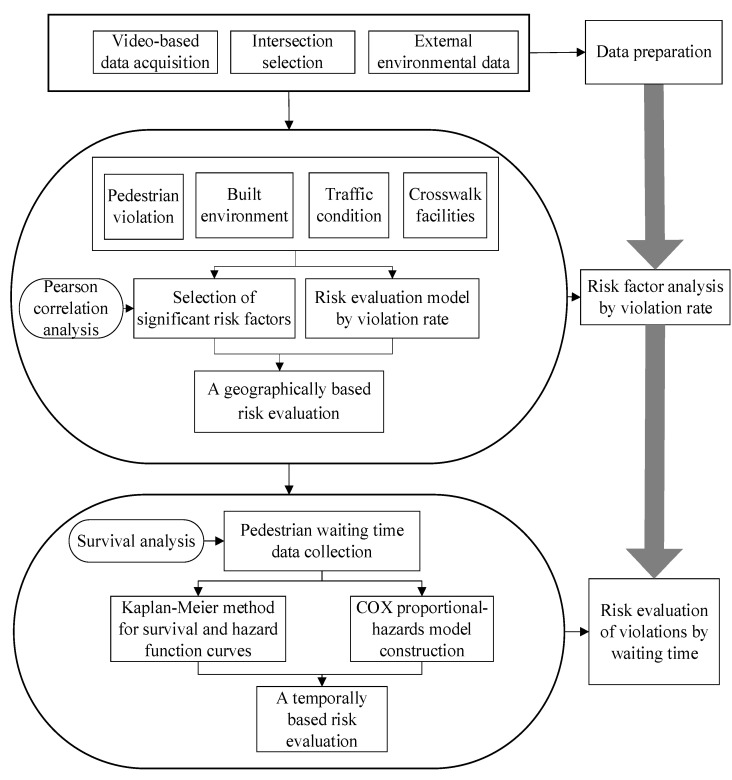
Research framework.

**Figure 2 ijerph-19-14420-f002:**
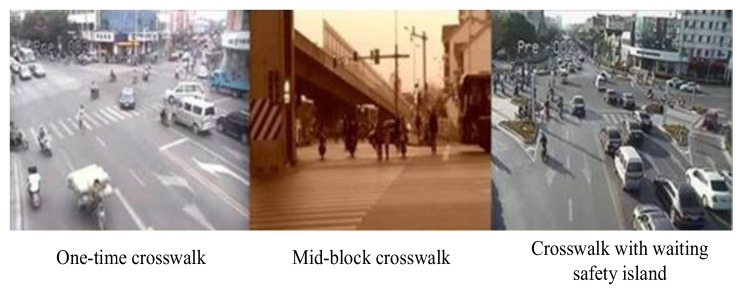
Three types of pedestrian crosswalk facilities at signal intersections.

**Figure 3 ijerph-19-14420-f003:**
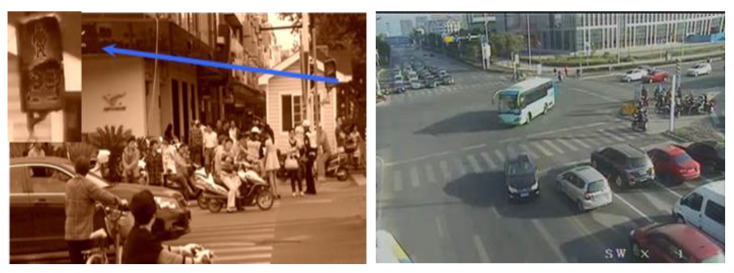
The process of data acquisition from surveillance videos of intersections.

**Figure 4 ijerph-19-14420-f004:**
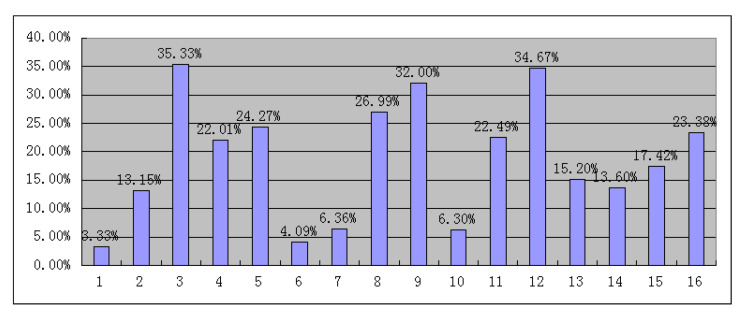
Rate of pedestrian violation at each studied intersection.

**Figure 5 ijerph-19-14420-f005:**
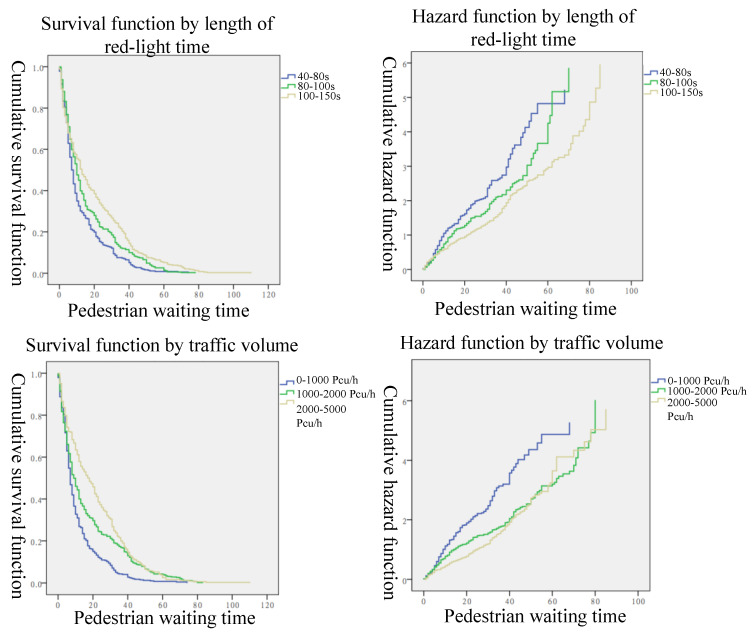
The survival and hazard function curves by four different lengths of red-light time and traffic volume.

**Figure 6 ijerph-19-14420-f006:**
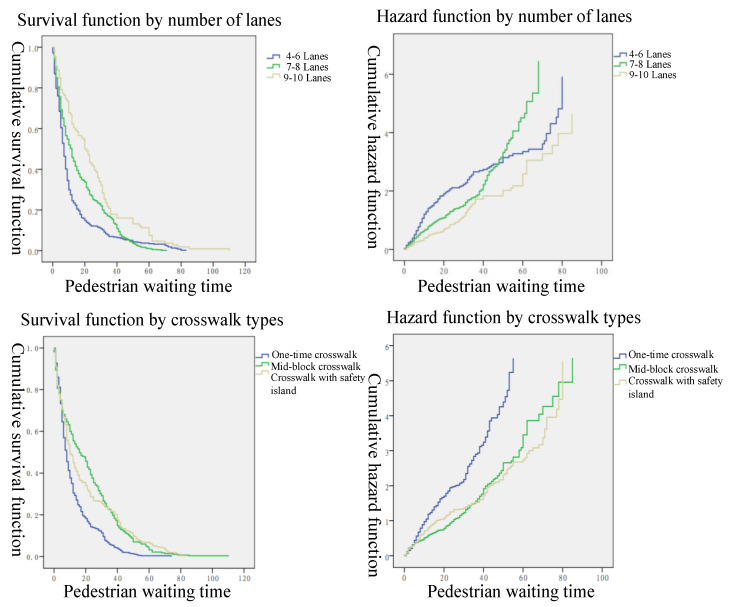
The survival and hazard function curves by different numbers of lanes and crosswalk types.

**Table 1 ijerph-19-14420-t001:** Variable assignment of risk factors.

Risk Factors	Investigation Parameters	Category	Variable Assignment
Traffic conditions	Pedestrian waiting time for red light	Continuous variable	Survey confirmation
	Traffic volume at crosswalk	Discrete variable	Video statistical acquisition
	Whether to set a red-light countdown	Nominal variable	Yes = 1, no = 0
Built environment	Number of lanes	Continuous variable	Survey confirmation
	Land use type around the crosswalk	Nominal variable	Residential area = 1, mixed area = 2, and commercial area = 3
	Crosswalk isolation degree	Nominal variable	None = 0, fence = 1, and green belt = 2
	Turn right traffic flow	Discrete variable	Video statistical acquisition
	Signal clearing time	Continuous variable	Survey confirmation
Crosswalk facilities	Crossing distance	Continuous variable	Survey confirmation
	Crosswalk type	Nominal variable	One-time crosswalk = 1, mid-block crosswalk = 2, and crosswalk with refuge island = 3

**Table 2 ijerph-19-14420-t002:** Correlation analysis between risk factors and violation rates.

		Crossing Distance	Number of Lanes	Traffic Volume at Crosswalk	Signal Clearing Time	Length of Red-Light Time
Violation rate	Pearson correlation coefficient	0.57	−0.028	0.551	0.228	0.668
	t-value significance test	0.037	0.541	0.05	0.479	−0.028
	Sample size	16	16	16	16	16
		**Land Use Type Around Crosswalk**	**Crosswalk Isolation Degree**	**Whether to Set Red-Light Countdown**	**Turn Right Traffic Flow**	**Crosswalk Type**
Violation rate	Pearson correlation coefficient	−0.337	−0.228	0.43	0.479	0.79
	t-value significance test	0.396	0.678	0.35	0.178	0.018
	Sample size	16	16	16	16	16

**Table 3 ijerph-19-14420-t003:** Geographical risk model coefficients and t-values.

	Coefficient	Significance
Constants	4.579	0.081
Crossing distance	−3.662	0.037
Traffic volume	−5.097	0.05
Red-light time	2.51	0.028
Crosswalk type	−10.544	0.018

**Table 4 ijerph-19-14420-t004:** Backward elimination step for temporal risk factors.

Variable		Regression Coefficient	Standard Error of Regression Coefficient	Ward Statistical Values	Relative Hazard
Step 1	Crosswalk type	−0.140	0.066	3.134	0.786
	Gender	−0.084	0.093	3.936	0.832
	Age	−0.010	0.055	0.033	0.990
	Red-light time	−0.164	0.004	0.687	1.004
	Crosswalk distance	0.231	0.003	1.212	0.745
	Traffic volume	0.410	0.021	1.471	1.000
	Number of lanes	0.128	0.064	0.015	1.008
Step 2	Crosswalk type	−0.141	0.066	3.235	0.786
	Age	−0.131	0.092	3.928	0.833
	Red-light time	−0.180	0.055	0.036	0.990
	Crosswalk distance	0.237	0.006	2.625	0.645
	Traffic volume	0.423	0.04	0.973	1.004
	Number of lanes	0.130	0.03	2.334	1.000
Step 3	Crosswalk type	−0.137	0.029	2.213	0.235
	Red-light time	−0.152	0.073	1.223	0.412
	Crosswalk distance	0.226	0.061	3.472	0.462
	Traffic volume	−0.317	0.015	0.751	0.819
	Number of lanes	0.213	0.012	2.362	0.916
Step 4	Crosswalk type	−0.120	0.066	3.250	0.786
	Red-light time	−0.183	0.092	3.893	0.834
	Traffic volume	−0.434	0.004	0.993	1.004
	Number of lanes	0.144	0.000	1.386	1.000

## Data Availability

All data used during the study can be download for free at https://github.com/xingjiping/Data-of-waiting-time-for-pedestrian-violation (26 September 2022).

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
