# Peer review of "A Geographical and Temporal Risk Evaluation Method for Red-Light Violations by Pedestrians at Signalized Intersections: Analysis and Results of Suzhou, China"

_ijerph, 2022, doi:10.3390/ijerph192114420_

Round 1

Reviewer 1 Report

The reviewed paper is focused on the proposal of a geographical and temporal risk evaluation method for pedestrian red-light violations by combining real survey data and video-based acquisition when applying both (1) the Pearson correlation analysis in order to extract the significant affecting criteria from traffic environment, built environment, and crosswalk facilities, and (2) the survival analysis method to determine the risk of pedestrian violation in different scenarios as time passes by in Suzhou, China. Following the afforementioned, the adequate methods of examination are applied in the research conducted. Furthermore, the authors have elaborated quite in-depth and meticulous analysis of the existing literature (see the Literature review section) in the topic addressed, and they have adhered all the required formal as well as content-related publishing conditions stipulated by the "International Journal of Environmental Research and Public Health". The level of English in the text is appropriate and paper structure being well compiled. All the findings and conclusions appear to be justified and properly formulated.

However,  despite the relatively high quality odf the research conducted, I am still able to detect some minor deficiencies. Please see them as follows:

1.) The structure of the abstract is to be slightly modified. Please meet the ensuing specified structure and, in addition, emphasize the determined crucial objective of the research in detail. "a) brief description of all the manuscript sections; b) defining the crucial manuscript objective (aim); c) brief outlining the major distinctions of the research conducted in comparison with other previously published works. The manuscript crucial objective is not defined explicitly. Moreover, in its current form, the abstract section is too short, thus it needs to be expanded."

2.) Discussion section, encompassing all the major findings out of the research conducted when summarizing all the quantified relevant outcomes (by incorporating some additional summary computational tables, charts, schemes, figures, diagrams), needs to be elaborated.

3.) The present list of up-to-date references (38 sources precisely) is relatively extensive, and yet, in order to enhance the comprehensive scientific soundness of the research conducted, I recommend adding even more topic-related (WoS, Scopus-indexed) internationally-wide literature sources to portray the subject discussed even from broader perspective. See some of them as follows:

- A multivariate method for evaluating safety from conflict extremes in real time. Analytic Methods in Accident Research, 2022, 36. DOI: 10.1016/j.amar.2022.100244.

- Methodology for Estimating the Effect of Traffic Flow Management on Fuel Consumption and CO2 Production: A Case Study of Celje, Slovenia. Energies, 2021, 14(6), Art. no. 1673. DOI: 10.3390/en14061673.

- Algorithm for Creating Optimized Green Corridor for Emergency Vehicles with Minimum Possible Disturbance in Traffic. LOGI – Scientific Journal on Transport and Logistics, 2022, 13 (1), pp. 84-95. DOI: 10.2478/logi-2022-0008.

- A functional approach for characterizing safety risk of signalized intersections at the movement level: An exploratory analysis. Accident Analysis and Prevention, 2021, 163, Art. no. 106446. DOI: 10.1016/j.aap.2021.106446.

- Estimates of Economic Cost of Congestion Travel Time Delay between Onne-Seaport and Eleme-Junction Traffic Corridor. LOGI – Scientific Journal on Transport and Logistics, 2020, 11(2), pp. 33-43. DOI: 10.2478/logi-2020-0013.

- Proposal of a Roundabout Solution within a Particular Traffic Operation. Open Engineering, 2016, 6(1), pp. 441-445. DOI: 10.1515/eng-2016-0066.

- Disaggregated traffic conditions and road crashes in urban signalized intersections. Journal of Safety Research, 2021, 77, pp. 202-211. DOI: 10.1016/j.jsr.2021.03.003.

- and so on

After a due revision of the above remarks and shortcomings, the manuscript will be ready for publishing. Good luck.

Author Response

Response to Comments of Reviewer 1

The reviewed paper is focused on the proposal of a geographical and temporal risk evaluation method for pedestrian red-light violations by combining real survey data and video-based acquisition when applying both (1) the Pearson correlation analysis in order to extract the significant affecting criteria from traffic environment, built environment, and crosswalk facilities, and (2) the survival analysis method to determine the risk of pedestrian violation in different scenarios as time passes by in Suzhou, China. Following the afforementioned, the adequate methods of examination are applied in the research conducted. Furthermore, the authors have elaborated quite in-depth and meticulous analysis of the existing literature (see the Literature review section) in the topic addressed, and they have adhered all the required formal as well as content-related publishing conditions stipulated by the "International Journal of Environmental Research and Public Health". The level of English in the text is appropriate and paper structure being well compiled. All the findings and conclusions appear to be justified and properly formulated. However,  despite the relatively high quality odf the research conducted, I am still able to detect some minor deficiencies. Please see them as follows.

Thank you for your encouragement and recognition of the research topic. Your comments and suggestions have been carefully investigated and are quite important for the amendments to this manuscript. Our detailed responses to your comments are provided as follows:

Comment 1-1:   The structure of the abstract is to be slightly modified. Please meet the ensuing specified structure and, in addition, emphasize the determined crucial objective of the research in detail. "a) brief description of all the manuscript sections; b) defining the crucial manuscript objective (aim); c) brief outlining the major distinctions of the research conducted in comparison with other previously published works. The manuscript crucial objective is not defined explicitly. Moreover, in its current form, the abstract section is too short, thus it needs to be expanded."

Response:         Thank you for your suggestion. We have realized that our abstract should be content of our abstract should be a bit more information. In the revised manuscript, the several sentences have been re-added, the new abstract is listed as follows:

“Red-light violations of pedestrian crossing at signal intersections is one of the key factors in pedes-trian traffic accidents. Even though there are various studies on pedestrian behavior and pedestrian traffic conflicts, few research focuses on the risk of different crosswalks for the violating pedestrian group. Due to the spatio-temporal nature of violation risk, this study proposes a geographical and temporal risk evaluation method for pedestrian red-light violations, which combines actual survey and video acquisition. First, in the geographical-based risk evaluation, the pedestrian violation rate in signal intersections is investigated by Pearson correlation analysis to extract the significant in-fluencing factors from traffic conditions, built environment, and crosswalk facilities. Second, in the temporal-based risk evaluation, the survival analysis method is developed to quantify the risk of pedestrian violation in different scenarios as time passes by. Finally, this study select 16 typical signalized intersections in Suzhou, China, with 881 pedestrian crosswalk violations from a total size of 4,586 pedestrians as survey cases. Results indicate that crossing distance, traffic volume on the crosswalk, red-light time, and crosswalk type variables all contribute to the effect of pedestrian vi-olation in geographical perspective, and the installation of waiting refuge islands has the most significant impact. From the temporal perspective, the increase in red-light time, number of lanes, and traffic volume have a mitigating effect on the violations with pedestrian waiting time increases. This study aims to provide a development-oriented path by proposing an analytical framework that reconsiders geographical and temporal risk factors of violation. The findings could help transport planners understand the effect of pedestrian violation related traffic risk and develop operational measures and crosswalk design schemes for controlling pedestrian violation occurring on local communities.”

Comment 1-2:   Discussion section, encompassing all the major findings out of the research conducted when summarizing all the quantified relevant outcomes (by incorporating some additional summary computational tables, charts, schemes, figures, diagrams), needs to be elaborated.

Response:         Thanks for your keen observation. We have re-added descriptions to some of the figures and tables in the revised manuscript. Furthermore, we have carefully proofread the whole manuscript and double-checked for typos and mistakes.

Comment 1-3:   The present list of up-to-date references (38 sources precisely) is relatively extensive, and yet, in order to enhance the comprehensive scientific soundness of the research conducted, I recommend adding even more topic-related (WoS, Scopus-indexed) internationally-wide literature sources to portray the subject discussed even from broader perspective. - A multivariate method for evaluating safety from conflict extremes in real time. Analytic Methods in Accident Research, 2022, 36. DOI: 10.1016/j.amar.2022.100244.- Methodology for Estimating the Effect of Traffic Flow Management on Fuel Consumption and CO2 Production: A Case Study of Celje, Slovenia. Energies, 2021, 14(6), Art. no. 1673. DOI: 10.3390/en14061673.- Algorithm for Creating Optimized Green Corridor for Emergency Vehicles with Minimum Possible Disturbance in Traffic. LOGI – Scientific Journal on Transport and Logistics, 2022, 13 (1), pp. 84-95. DOI: 10.2478/logi-2022-0008- A functional approach for characterizing safety risk of signalized intersections at the movement level: An exploratory analysis. Accident Analysis and Prevention, 2021, 163, Art. no. 106446. DOI: 10.1016/j.aap.2021.106446.- Estimates of Economic Cost of Congestion Travel Time Delay between Onne-Seaport and Eleme-Junction Traffic Corridor. LOGI – Scientific Journal on Transport and Logistics, 2020, 11(2), pp. 33-43. DOI: 10.2478/logi-2020-0013.- Proposal of a Roundabout Solution within a Particular Traffic Operation. Open Engineering, 2016, 6(1), pp. 441-445. DOI: 10.1515/eng-2016-0066.-Disaggregated traffic conditions and road crashes in urban signalized intersections. Journal of Safety Research, 2021, 77, pp. 202-211. DOI: 10.1016/j.jsr.2021.03.003.

Response:         Thank you for your suggestion. We have reconsidered our research topic and added these references in the revised manuscript.

Once more, we sincerely thank the reviewers for the helpful comments provided, which have significantly improved the work presented in this paper.

Reviewer 2 Report

This study evaluates geographical and temporal-based pedestrian red-light violation risk. The Pearson correlation analysis and survival analysis methods are applied to determine the significant influencing factors of the intersection violation rate. Influencing factors of pedestrian waiting time are identified using COX proportional-hazards model. Data has been collected using video and on-site surveys. The topic is research worthy and very contemporary. The paper is also well-written and structured.

 The main strength of this paper is the description of the methodology used. And the main weakness or limitation is the selection of factors for crossing violation evaluation. This study considered all pedestrians under the same platform. It only deals with a few geographical characteristics and temporal factors. Why not include demographic or physical characteristics say, age, gender, trip purpose, trip distance, time of the trip, etc.? Philological or mental effects are also a big concern.  Need to address this limitation and at least recommend it for future research.

 Few more comments:

 Abstract: It is suggested to incorporate a sentence on findings, at least the name of significant factors.

Literature review: Well-written LR. Pointed out the gaps critically and justify the study context. However, some sentences are too long. Better to break those. The last sentence should be in the Data collection part.

 Crossing characteristics in commercial districts and residential areas are different (page 4). Whether it is a hypothesis or backed by any reference, needs to be mentioned.

 Some references are missing in some strategic locations. E.g., Section 3.3?

 The methodology of analysis is clear and well-written. However, there should have a method for Data validation. How the quality of data has been maintained? How to control the observer’s errors? Nothing about observers or surveyors.

 A case study could be renamed “Analysis and Results”.

In this section, a few sentences could be considered Redundant, and or transfer to the methodology section".

There should have a strong correlation between, crossing distance and the Number of lanes. Even, in The Temporal Risk Factors analysis, the crossing distance is determined by the number of lanes, and the width of the lane is considered fixed (last para, page 11). Hence, both should provide the same results when the correlation is drawn with the violation rate. But table 2 shows different results. How do you explain this?

 The same observation is for traffic volume and signal clearing time.

 Page 11, 3rd Para. Repetition. Better to shift to the methodology section.

 Give a sample example to explain equations 9 & 10.

 Discussion of the findings is minuscule. It should be backed by reference.

Conclusions: Some specific countermeasures should be suggested. A few more research limitations and directions may improve the research value of the paper. 

Author Response

Response to Comments of Reviewer 2

This study evaluates geographical and temporal-based pedestrian red-light violation risk. The Pearson correlation analysis and survival analysis methods are applied to determine the significant influencing factors of the intersection violation rate. Influencing factors of pedestrian waiting time are identified using COX proportional-hazards model. Data has been collected using video and on-site surveys. The topic is research worthy and very contemporary. The paper is also well-written and structured.

Thank you for your encouragement and recognition of the research topic. Your comments and suggestions have been carefully investigated and are quite important for the amendments of this manuscript. Our detailed responses to your comments are provided as follows:

Comment 2-1:   The main strength of this paper is the description of the methodology used. And the main weakness or limitation is the selection of factors for crossing violation evaluation. This study considered all pedestrians under the same platform. It only deals with a few geographical characteristics and temporal factors. Why not include demographic or physical characteristics say, age, gender, trip purpose, trip distance, time of the trip, etc.? Philological or mental effects are also a big concern.  Need to address this limitation and at least recommend it for future research.

Response:          Thank you for your highly valuable comment. The above factors proposed by this question are critical to the risk evaluation of pedestrian violation, and we admit that our research limitation is unclearly expressed and difficult to understand in the manuscript. Actually, our study aims to evaluate the impact of external environmental factors on crosswalks at the intersection leading to the risk of violation occurring for pedestrians. The entire crosswalk at an intersection in one direction is the object of our study. The group of the pedestrian at a crosswalk is analyzed as a whole, and we select the rate of pedestrian violation as the criteria for risk evaluation. Hence, in geographical-based risk evaluation, the built environment, traffic condition, and crosswalk facilities from each different crosswalk are viewed as three core types of external environmental factors in our manuscript. And the trends of selected external risk factors in the temporal perspective with the waiting time of pedestrians increases are analyzed. Furthermore, limited by the difficulty of obtaining characteristics of the travel purpose and trip distance of the violating pedestrians with our video survey method, we did not employ these factors into our consideration. And the philological or mental effects of pedestrian violations have been considered in previous studies from the perspective of the individual pedestrians themselves. In further research, by combining the questionnaire, we could explore the entire pedestrian crosswalk perspective to further investigate the effect of these factors. In the revised manuscript, we have re-summarized the study motivation (in Section 1) and research limitations (in Section 6) as follows: 

“In this context, extensive efforts have been undertaken to investigate the influential factors of pedestrian violation crossing on red-light time. Among them, present studies can be divided into internal human factors and external environment by the differences in the purpose of the investigation. The internal human factors mainly consider the effects of age (Trpković et al., 2017), gender (Ni et al., 2017), gap acceptance (Demiroz et al., 2015; Ma et al., 2020), mental effects (Useche and Llamazares, 2022) and crossing behavior selection as an individual or group of pedestrians (Ma et al., 2020). And external environment factors includes built environment features (Congiu et al., 2019), traffic conditions (Shaaban et al., 2018), the length of red-light time (Aghabayk et al., 2021), time of the trip (Liu et al., 2022), social characteristics (Raoniar et al., 2022) and road crosswalk facilities (Alonso et al., 2021; Shaaban et al., 2018). It is a decision-making process of physical and mental for pedestrians from the moment they arrive at a particular signalized intersection to the moment they are ready to violate the crossing. As signal intersection is a complex traffic environment in urban transportation, the diversity of inherent personal characteristics and extrinsic intersection attributes can simultaneously affect pedestrian violations. This process has a high degree of uncertainty, different pedestrians tend to make different crossing strategies under different corsswalk scenarios and waiting times.

However, most of the present studies on the risk evaluation of pedestrian violations have focused on the perspective of individual pedestrian characteristics, and discussed the risk of illegal crossing from the perspective of pedestrians themselves. Fewer studies have discussed the risk of pedestrian crossing violations occurring from the effect of the entire crossing crosswalk. Limited by the randomness and variability of pedestrians arriving at the crosswalk, measures to reduce the risk of violation from the pedestrian's perspective are uncertain. Instead, a whole crosswalk perspective can provide some suggestions for stable improvements based on the external environment. As such, it is crucial to consider geographical characteristics and temporal trends of selected external factors in the risk evaluation of violation pedestrians.”

“In future research, with the help of the questionnaire, we could further extend crosswalk perspective to investigate the effect of physical characteristics, trip purpose, trip distance, and trip time of pedestrian.”

Reference here

Aghabayk, K., Esmailpour, J., Jafari, A., Shiwakoti, N., 2021. Observational-based study to explore pedestrian crossing behaviors at signalized and unsignalized crosswalks. Accident Analysis and Prevention 151, 12-34.

Alonso, F., Oviedo-Trespalacios, O., Gene-Morales, J., Useche, S.A., 2021. Assessing risky and protective behaviors among pedestrians in the Dominican Republic: New evidence about pedestrian safety in the Caribbean. Journal of Transport & Health 22, 101145.

Congiu, T., Sotgiu, G., Castiglia, P., Azara, A., Dettori, M., 2019. Built environment features and pedestrian accidents: An italian retrospective study. Sustainability 11, 121-134.

Demiroz, Y.I., Onelcin, P., Alver, Y., 2015. Illegal road crossing behavior of pedestrians at overpass locations: Factors affecting gap acceptance, crossing times and overpass use. Accident Analysis and Prevention 80, 220-228.

Liu, Y., Alsaleh, R., Sayed, T., 2022. Modeling pedestrian temporal violations at signalized crosswalks: A random Intercept parametric survival model. Transportation Research Record 2676, 707-720.

Ma, Y., Lu, S., Zhang, Y., 2020. Analysis on illegal crossing behavior of pedestrians at signalized intersections based on bayesian network. Journal of Advanced Transportation 2020, 23-45.

Ni, Y., Cao, Y., Li, K., 2017. Pedestrians' safety perception at signalized intersections in Shanghai. Transportation Research Procedia 25, 1960-1968.

Qin, W., Ji, X., Liang, F., 2020. Estimation of urban arterial travel time distribution considering link correlations. Transportmetrica A: Transport Science 16, 1429-1458.

Raoniar, R., Maqbool, S., Pathak, A., Chugh, M., Maurya, A.K., 2022. Hazard-based duration approach for understanding pedestrian crossing risk exposure at signalised intersection crosswalks - A case study of Kolkata, India. Transportation Research Part F-Traffic Psychology and Behaviour 85, 47-68.

Shaaban, K., Muley, D., Mohammed, A., 2018. Analysis of illegal pedestrian crossing behavior on a major divided arterial road. Transportation Research Part F-Traffic Psychology and Behaviour 54, 124-137.

Sobreira, L.T.P., Cunto, F., 2021. Disaggregated traffic conditions and road crashes in urban signalized intersections. Journal of Safety Research 77, 202-211.

Trpković, A., Milenković, M., Vujanić, M., Stanić, B., Glavić, D., 2017. The crossing speed of elderly pedestrians. Promet-Traffic & Transportation 29, 175-183.

Useche, S.A., Llamazares, F.J., 2022. The guilty, the unlucky, or the unaware? Assessing self-reported behavioral contributors and attributions on pedestrian crashes through structural equation modeling and mixed methods. Journal of Safety Research 82, 329-341.

Yang, D., Ozbay, K., Xie, K., Yang, H., Zuo, F., 2021. A functional approach for characterizing safety risk of signalized intersections at the movement level: An exploratory analysis. Accident Analysis and Prevention 163.

Comment 2-2:   Abstract: It is suggested to incorporate a sentence on findings, at least the name of significant factors.

Response:          Thank you for your insightful comments, following your suggestions, we have re-added several sentences on the influencing factors we identified in the abstract in the revised manuscript, which is listed as follows.

“Results indicate that crossing distance, traffic volume on the crosswalk, red-light time, and crosswalk type variables all contribute to the effect of pedestrian violation in a geographical perspective, and the installation of waiting refuge islands has the most significant impact. From the temporal perspective, the increase in red-light time, number of lanes, and traffic volume have a mitigating effect on the violations with pedestrian waiting time increases. This study aims to provide a development-oriented path by proposing an analytical framework that reconsiders geographical and temporal risk factors of violation. The findings could help transport planners understand the effect of pedestrian violation related traffic risk and develop operational measures and crosswalk design schemes for controlling pedestrian violation occurring on local communities.”

Comment 2-3:   Literature review: Well-written LR. Pointed out the gaps critically and justify the study context. However, some sentences are too long. Better to break those. The last sentence should be in the Data collection part.

Response:          Thank you for the insightful comment. In the revised manuscript, we have invited a professional English editor (native speaker) to carefully proofread and revise the paper, and some long sentences have been broken. We believe that the English writing of this paper is largely polished and refined. Furthermore, following your suggestions, we have re-added several sentences to illustrate the data collection, which is summarized as follows:

“In general, previous studies on the geographical risk of pedestrian violations have been performed mainly from the perspective of the pedestrians themselves and analyzed the effect level of pedestrian accidents or fatalities with the help of macroscopic data (such as socio-economic development, demographic characteristics, and urban road network), or one specific factor data (Cheng et al., 2022; Yang et al., 2021a; Yang et al., 2022). Different types of pedestrians present different crossing characteristics under different research scenarios in terms of travel patterns and traffic activities. However, fewer studies have focused on the entire crosswalk at intersections to analyze the risk to pedestrians before an accident, which can help to propose improvements to mitigate the risk of pedestrian violations from the perspective of crosswalks. And this topic is crucial to improve the safe environment at intersections.

In summary, benefit from pedestrian waiting time data collected using interviews, questionnaires, video trajectory recording, etc (Raoniar et al., 2022; Useche et al., 2020; Useche et al., 2022). These studies have pointed out the maximum waiting times for pedestrian crosswalk violations and described the changing trends in the risk of pedestrians from pedestrians arriving at crosswalks to the occurrence of the violation. However, the aforementioned studies did not reflect the effects of various external factors on the risk of pedestrian violation involved. The impact of different levels of external factors on the temporal risk of pedestrian violations is of different degrees. For example, the waiting time for pedestrians under different crossing facilities is different.

Reference here

Raoniar, R., Maqbool, S., Pathak, A., Chugh, M., Maurya, A.K., 2022. Hazard-based duration approach for understanding pedestrian crossing risk exposure at signalised intersection crosswalks - A case study of Kolkata, India. Transportation Research Part F-Traffic Psychology and Behaviour 85, 47-68.

Useche, S.A., Alonso, F., Montoro, L., 2020. Validation of the Walking Behavior Questionnaire (WBQ): A tool for measuring risky and safe walking under a behavioral perspective. Journal of Transport & Health 18, 100899.

Useche, S.A., Gonzalez-Marin, A., Faus, M., Alonso, F., 2022. Environmentally friendly, but behaviorally complex? A systematic review of e-scooter riders’ psychosocial risk features. PLoS ONE 17.

Yang, C., Chen, M., Yuan, Q., 2021a. The application of XGBoost and SHAP to examining the factors in freight truck-related crashes: An exploratory analysis. Accident Analysis & Prevention 158, 106153.

Yang, Z., Chen, X., Pan, R., Yuan, Q., 2022. Exploring location factors of logistics facilities from a spatiotemporal perspective: A case study from Shanghai. Journal of Transport Geography 100, 103318.

Cheng, Z., Zhang, L., Zhang, Y., Wang, S., Huang, W., 2022. A systematic approach for evaluating spatiotemporal characteristics of traffic violations and crashes at road intersections: an empirical study. Transportmetrica A: Transport Science, 1-23.

Comment 2-4:   Crossing characteristics in commercial districts and residential areas are different (page 4). Whether it is a hypothesis or backed by any reference, needs to be mentioned.

Response:         Thanks for your keen observation. We apologize for this omission, and some explanations have been re-added in our revised manuscript, which is listed as follows.

“It should be noted that in the division of land use types, based on the percentage of building types within 100 meters radius around the crosswalk at the intersection, land use type around the crosswalk can be divided into the residential area, commercial area, and mixed area. For example, if a supermarket or residential community appears near the intersection, it can be classified as a commercial area or residential area respectively, otherwise, it is regarded as a mixed zone (Jang et al., 2013; Rankavat and Tiwari, 2016).”

Reference here

  1. Jang, S. H. Park, S. Kang, K. H. Song, S. Kang, and S. Chung, "Evaluation of pedestrian safety pedestrian crash hot spots and risk factors for injury severity," Transportation Research Record, no. 2393, pp. 104-116, 2013 2013.
  2. Rankavat and G. Tiwari, "Pedestrians risk perception of traffic crash and built environment features - Delhi, India," Safety Science, vol. 87, pp. 1-7, 2016.

Comment 2-5:   Some references are missing in some strategic locations. E.g., Section 3.3?

Response:          Thank you for your highly valuable comment. We have realized some of the original information and new findings are missing references for introduction and comparison respectively. In the revised manuscript, we have re-added some references and new explanations, and part of them are listed as follows.

Reference here

  1. S. Pulugurtha and V. R. Sambhara, "Pedestrian crash estimation models for signalized intersections," Accident Analysis & Prevention, vol. 43, no. 1, pp. 439-446, 2011.
  2. T. P. Sobreira and F. Cunto, "Disaggregated traffic conditions and road crashes in urban signalized intersections," Journal of Safety Research, vol. 77, pp. 202-211, 2021.
  3. Cheng, Z. Zu, J. Lu, and Y. Li, "Exploring the Effect of Driving Factors on Traffic Crash Risk among Intoxicated Drivers: A Case Study in Wujiang," International Journal of Environmental Research and Public Health, 14, pp. 23-45, 2019
  4. Yang, K. Ozbay, K. Xie, H. Yang, and F. Zuo, "A functional approach for characterizing safety risk of signalized intersections at the movement level: An exploratory analysis," Accident Analysis and Prevention, vol. 163, 2021.

Comment 2-6:   The methodology of analysis is clear and well-written. However, there should have a method for data validation. How the quality of data has been maintained? How to control the observer’s errors? Nothing about observers or surveyors.

Response:          Thank you for the recognition and insightful comments. In the application of data validation methods, the t-value statistical test has been used for selecting the significant variables in the Pearson correlation analysis in geographical risk evaluation, and the Backward elimination test has been used to filter the variables correlated with pedestrian waiting time in temporal risk evaluation. Just as this part of the results does not have an impact on the conclusions of our study, we did not put this processing into the original manuscript. In our revised manuscript, we have re-explained this part, which is listed as follows.

“According to the requirement of a t-value significance test, the t-value of significant influencing factors should be less than 0.05 (Schober et al., 2018).

Furthermore, for different types of pedestrians, we further classify them into different ages (Teenager=1, Adult=2, and Elderly=3 ) and gender (Male=1, and Female=2) based on their physical features in the video.

Then, we also need to verify whether these factors play a critical role in our pedestrian waiting time experiments. As for avoiding the omission of potentially valuable variables, the Backward elimination method is applied to gradually remove the insignificant covariates (Sauerbrei et al., 1992). As shown in Table 4, since the factors of age and gender, and waiting time are not significant, they are eliminated in the first and second steps, respectively.

Table 4. Backward elimination step for temporal risk factors.

Variable

Regression

coefficient

Standard error of regression coefficient

Ward statistical values

Relative hazard

Step 1

Crosswalk type

-0.140

0.066

3.134

0.786

Gender

-0.084

0.093

3.936

0.832

Age

-0.010

0.055

0.033

0.990

Red-light time

-0.164

0.004

0.687

1.004

Crosswalk distance

0.231

0.003

1.212

0.745

Traffic volume

0.410

0.021

1.471

1.000

Number of lanes

0.128

0.064

0.015

1.008

Step 2

Crosswalk type

-0.141

0.066

3.235

0.786

Age

-0.131

0.092

3.928

0.833

Red-light time

-0.180

0.055

0.036

0.990

Crosswalk distance

0.237

0.006

2.625

0.645

Traffic volume

0.423

0.04

0.973

1.004

Number of lanes

0.130

0.03

2.334

1.000

Step3

Crosswalk type

-0.137

0.029

2.213

0.235

Red-light time

-0.152

0.073

1.223

0.412

Crosswalk distance

0.226

0.061

3.472

0.462

Traffic volume

-0.317

0.015

0.751

0.819

Number of lanes

0.213

0.012

2.362

0.916

Step4

Crosswalk type

-0.120

0.066

3.250

0.786

Red-light time

-0.183

0.092

3.893

0.834

Traffic volume

-0.434

0.004

0.993

1.004

Number of lanes

0.144

0.000

1.386

1.000

Furthermore, as for data quality maintenance and surveyor error control in our study, our data collection can be divided into fixed external factor data (i.e. number of lanes, red-light time, traffic volume on crosswalk and crosswalk type) and variable pedestrian waiting time data. In fixed external data collection, we employed a field survey to collect data from 16 intersections. And these data have been published on the GitHub website and the appendix, which can be easily validated in the following studies. In the collection of variable waiting time data, we applied a video survey method to capture pedestrian waiting times, where about 4,800 pedestrian crossing processes were recorded based on government service projects and accomplished by six different investigators. We acknowledge that there is some error in the data collection of violation waiting time. Particularly, the judgment of whether a violation has occurred after a period of pedestrian waiting, and the recording of the start time of multiple pedestrians arriving at a crosswalk simultaneously. In the former error consideration, we divide the waiting time into censored data and non-censored data to mitigate the effect of violation waiting time being mistaken for normal passing waiting time, which is an important method in survival analysis. In the latter, we put these error considerations into the section on research limitations. And all of them have been re-added in the revised manuscript, which is listed as follows.

“Although this study used a sample that was representative of pedestrians and the essential statistical parameters, and theoretical assumptions were coherently achieved during the data analysis, there still have some limitations and potential biasing sources in collection of violation waiting time in that we must be acknowledged. Particularly, the judgment of whether a violation has occurred after a period of pedestrian waiting, and the recording of the start time of multiple pedestrians arriving at a crosswalk simultaneously. However, this does not inhibit the influence of common method biases and variance, mainly if sensitive issues related to age, gender, and violation criteria from pedestrians are addressed, and these do not prejudice our discussion of the external risk factors for crosswalks.”

Reference here

  1. Schober, C. Boer, and L. A. Schwarte, "Correlation coefficients: appropriate use and interpretation," Anesthesia and Analgesia, vol. 126, no. 5, pp. 1763-1768, 2018.
  2. Sauerbrei and M. Schumacher, "A boostrap resampling procedure for model-building application to the COX regression model," Statistics in Medicine, vol. 11, no. 16, pp. 2093-2109, 1992.

Comment 2-7:   A case study could be renamed “Analysis and Results”.

Response:         Thank you for the insightful comment. We have changed the original title to “A geographical and temporal risk evaluation method for red-light violations by pedestrians at signalized intersections: Analysis and results of Suzhou, China” in the revised manuscript.

Comment 2-8:   In this section, a few sentences could be considered Redundant, and or transfer to the methodology section".There should have a strong correlation between, crossing distance and the Number of lanes. Even, in The Temporal Risk Factors analysis, the crossing distance is determined by the number of lanes, and the width of the lane is considered fixed (last para, page 11). Hence, both should provide the same results when the correlation is drawn with the violation rate. But table 2 shows different results. How do you explain this?

Response:         Thanks for your keen observation. This typo has been modified in the revised manuscript. We apologize for this confusing expression and have rewritten the expression in our manuscript.

“Furthermore, combining the factors selected by similar studies, the factor of age, gender, and the number of lanes also cannot be neglected (Liu et al., 2022). Hence, age, gender, red-light time, traffic volume, number of lanes, and cross-walk type are considered initial temporal risk factors.

Note that the number of lanes does not equate to the crosswalk distance, which is dependent on whether the crosswalk is set up with crossing refuge safety islands and road central barrier or not. And unlike in the geographical risk analysis which is influenced by the violation rate of crosswalks, there is some co-linearity problem between each influence factor. The factor analysis in the temporal risk takes the violation waiting time of each pedestrian as the dependent variable and analyzes the variation of waiting time of different pedestrians in different external environments. Therefore, it is acceptable that there are differences in the factors extracted for geographic and temporal risk analysis.”

Reference here

Liu, Y., Alsaleh, R., Sayed, T., 2022. Modeling pedestrian temporal violations at signalized crosswalks: A random Intercept parametric survival model. Transportation Research Record 2676, 707-720.

Comment 2-9:   The same observation is for traffic volume and signal clearing time.

Response:          Thank you for your highly valuable comment. In the geographic risk evaluation, the traffic volume is denoted as the number of vehicles driving across the crosswalk. The signal clearing time represents the transition signal for the green to red switch in the crosswalk and is used for pedestrians who have entered the crosswalk during the green time to complete the crossing process. During pedestrian signal clearing time, pedestrians are not allowed to re-enter the crosswalk. Table 2 is based on t-values to reflect their significance levels. The t-value of traffic volume is 0.05, which is compliant with the requirements and retains. On the contrary, the t-value of signal clearing time is greater than 0.05, and it was removed.

Comment 2-10:  Page 11, 3rd Para. Repetition. Better to shift to the methodology section.

Response:         Thank you for the insightful comments, following your suggestions, we have moved this paragraph to the methodology section in the revised manuscript.

Comment 2-11:  Give a sample example to explain equations 9 & 10.

Response:         Thank you for the insightful comment. In the revised manuscript, we have added the notation of equations 9 and 10. Furthermore, a sample example has been re-added, which is listed as follows.

“Based on the SPSS statistical software, the regression coefficients of the model are obtained in table 4, and our proposed COX proportional-hazards model can be expressed as equation (9):

   (9)

where t is survival time, and it denotes the waiting time before pedestrian violations,  denote basis hazard function, it is affected by external uncertainties, such as weather, local travel culture, etc (Sauerbrei and Schumacher, 1992). And  represent the value of the calculated risk value for pedestrian violations in time t. Furthermore, the variables of lane denote the number of lanes, the variable of redtime is the length of red-light signal time, crosswalk type represents the value of different crosswalk types, and trafficvolume is the value of the classified traffic volume category.

Due to the uncertainty , it is a nonparametric model. We usually apply a prognostic index (PI) to replace the calculation of fixed risk. The PI for the temporal risk of pedestrians is as follows:

 In equation (10), the variable coefficient reflects the degree of influence of each risk factor. When the coefficient is positive, it means that the hazard function decreased with the increase of the level of influencing factor. When the coefficient is negative, it means that the risk of pedestrian increase with the increase of the level of influencing factor. For example, in the specific traffic scenario of a crosswalk at an intersection, the value of PI can be obtained by inputting the number of lanes, with the category of red-light time, traffic volume, and crosswalk type. With a determined basis hazard , the temporal risk of pedestrian violation in time  can be acquired by .”

Reference here

Sauerbrei, W., Schumacher, M., 1992. A boostrap resampling procedure for model-building application to the COX regression model. Statistics in Medicine 11, 2093-2109.

Comment 2-12: Discussion of the findings is minuscule. It should be backed by reference.

Response:         Thanks for your keen observation. Following your suggestion, we have added some references and an introduction to argue our findings with other previous related studies for comparison and support in section 4.3 and section 4.4 revised manuscript.

Comment 2-13: Conclusions: Some specific countermeasures should be suggested. A few more research limitations and directions may improve the research value of the paper. 

Response:          Thank you for your insightful comments, following your suggestions, we have added research limitations and directions in section 6 in the revised manuscript, which is listed as follows.

“Based on our findings in the geographic and temporal risk analysis, for policy suggestions in improving the safety of pedestrians crossing at intersections, crosswalk planners should ideally install crossing refuge islands within the limitation of land use. And the set length of red-light time needs to be integrated comprehensively into different external environments.

Although this study used a sample that was representative of pedestrians and the essential statistical parameters, and theoretical assumptions were coherently achieved during the data analysis, there still have some limitations and potential biasing sources in the collection of violation waiting time that we must be acknowledged. Particularly, the judgment of whether a violation has occurred after a period of pedestrian waiting, and the recording of the start time of multiple pedestrians arriving at a crosswalk simultaneously. However, this does not inhibit the influence of common method biases and variance, mainly if sensitive issues related to age, gender, and violation criteria from pedestrians are addressed, and these do not prejudice our discussion of the external risk factors for crosswalks.

In future research, with the help of the questionnaire, we could further extend the crosswalk perspective to investigate the effect of physical characteristics, trip purpose, trip distance, and trip time of pedestrians. The evaluation of violation risk can be further analyzed by spatial geographical location, which includes the degree of impact on violation by different spatial distances between the intersection and between each intersection and the diversity of the built environment. Moreover, our study only utilized data from 16 signalized intersections, and consideration can be taken to increase the number of investigated intersections to comprehensively evaluate the relationship between the influencing factors in a large-scale urban road network.”

Once more, we sincerely thank the reviewers for the helpful comments provided, which have significantly improved the work presented in this paper.

Round 2

Reviewer 2 Report

Response to Data validation is not yet clear. 

Author Response

Comment 1-1: Response to Data validation is not yet clear. 

Response: Thanks for your insightful comment,and we agree with you that  the data validation is necessary in our research. Follow your suggestion, some introductions are re-added in revised manuscipt, which is listed as follow:

"Before conducting our work, we verified the reliability and validity of the data, and justified the data through discussions with experienced investigators."